# Gelsolin as a Potential Biomarker for Endoscopic Activity and Mucosal Healing in Ulcerative Colitis

**DOI:** 10.3390/biomedicines10040872

**Published:** 2022-04-09

**Authors:** Keiko Maeda, Masanao Nakamura, Takeshi Yamamura, Tsunaki Sawada, Eri Ishikawa, Akina Oishi, Shuji Ikegami, Naomi Kakushima, Kazuhiro Furukawa, Tadashi Iida, Yasuyuki Mizutani, Takuya Ishikawa, Eizaburo Ohno, Takashi Honda, Masatoshi Ishigami, Hiroki Kawashima

**Affiliations:** 1Department of Endoscopy, Graduate School of Medicine, Nagoya University, 65 Tsurumai-cho, Showa-ku, Nagoya 466-8550, Japan; t.sawada@med.nagoya-u.ac.jp; 2Department of Gastroenterology and Hepatology, Graduate School of Medicine, Nagoya University, 65 Tsurumai-cho, Showa-ku, Nagoya 466-8550, Japan; mnakamura@med.nagoya-u.ac.jp (M.N.); tyamamu@med.nagoya-u.ac.jp (T.Y.); erieri-i@med.nagoya-u.ac.jp (E.I.); baron@med.nagoya-u.ac.jp (A.O.); s.ikegami@med.nagoya-u.ac.jp (S.I.); nk202004@med.nagoya-u.ac.jp (N.K.); kazufuru@med.nagoya-u.ac.jp (K.F.); iidatyuw@med.nagoya-u.ac.jp (T.I.); y-mizu@med.nagoya-u.ac.jp (Y.M.); ishitaku@med.nagoya-u.ac.jp (T.I.); eono@med.nagoya-u.ac.jp (E.O.); honda@med.nagoya-u.ac.jp (T.H.); masaishi@med.nagoya-u.ac.jp (M.I.); h-kawa@med.nagoya-u.ac.jp (H.K.)

**Keywords:** biomarker, ulcerative colitis, gelsolin, mucosal healing

## Abstract

The therapeutic goal in ulcerative colitis is mucosal healing, which requires improved non-invasive biomarkers to evaluate disease activity. Gelsolin is associated with several autoimmune diseases, and here, we aimed to analyze its usefulness as a serological biomarker for clinical and endoscopic activities in ulcerative colitis. Patients with ulcerative colitis (*n* = 138) who had undergone blood tests and colonoscopy were included. Serum gelsolin was measured using enzyme-linked immunosorbent assay, and correlation between the gelsolin level and clinical and endoscopic activities was examined. The serum gelsolin level in patients with ulcerative colitis was significantly lower than that in healthy subjects, and it decreased in proportion to increasing Mayo score and Mayo endoscopic subscore. The area under the curve for correlation between clinical and endoscopic remission and serum gelsolin level was higher than that for C-reactive protein. Furthermore, in C-reactive protein-negative patients, the serum gelsolin level was lower in the active phase than in remission. Our findings indicate that the serum gelsolin level correlates with clinical and endoscopic activities in ulcerative colitis, has a higher sensitivity and specificity than C-reactive protein, and can detect mucosal healing, suggesting that gelsolin can be used as a biomarker for ulcerative colitis.

## 1. Introduction

Inflammatory bowel disease (IBD), represented by ulcerative colitis (UC) and Crohn’s disease, is a chronic inflammatory ailment of the gastrointestinal tract with an increasing incidence worldwide [1,2]. With recent advances in medical therapies, such as the development of immunomodulators and biologics, the goal of IBD treatment has shifted from alleviating clinical symptoms to achieving endoscopic mucosal healing. Mucosal healing reduces subsequent recurrence, surgery, and carcinogenesis rates, [3,4,5] and the concept of treat-to-target, aimed at achieving endoscopic mucosal healing, is being widely accepted [6,7]. The gold standard for determining disease activity and mucosal healing in IBD is endoscopy; however, this method is associated with physical, time, and economic burdens. Therefore, in clinical practice, non-invasive, and repeatable blood and stool-based biomarkers that reflect disease activity and mucosal healing are necessary. In addition to fecal markers such as fecal calprotectin and fecal immunochemical test, serum C-reactive protein (CRP) [8,9] and serum leucine rich glycoprotein (LRG) [10,11,12] have been reported as useful.

CRP is produced by hepatocytes in the acute phase upon IL-6 stimulation and is used as a biomarker for various inflammatory diseases. In terms of correlation with disease activity, CRP is associated with endoscopic activity in CD, but only with histologically severe inflammation in UC. Therefore, in UC, a low CRP does not necessarily mean the absence of endoscopic activity, which is problematic owing to its low sensitivity. Leucine-rich glycoprotein is also expressed on neutrophils, macrophages, and intestinal epithelial cells and is induced by interleukin (IL)-22, tumor necrosis factor-α, and IL-1 independent of IL-6. In CD, the LRG level is strongly correlated with disease activity, and in UC, it correlates with endoscopic activity, but more evidence is needed in this regard. Therefore, biomarkers that more accurately reflect clinical and endoscopic activities and predict mucosal healing than conventional markers are needed. The noninvasive assessment of accurate disease activity would enable optimal therapeutic choices and improve patient prognosis.

In this study, we performed proteasome analysis of colon mucus samples from patients with UC and focused on GSN, which is significantly downregulated in active UC compared with that in remission. Gelsolin (GSN) is an 82–84 kDa protein consisting of 730 amino acids organized into six homologous domains and expressed in both extracellular fluids and cytoplasm of most human cells [13,14]. GSN is a multifunctional protein; because of its strong effect on the cytoskeleton and inflammation-related biological processes, it shows potential as a biomarker for inflammation-associated medical conditions, such as for predicting illness severity, treatment efficacy, and clinical outcomes [15,16]. Reduced GSN level has been observed in patients with chronic autoimmune diseases such as rheumatoid arthritis and psoriasis [17,18]. GSN localization is also altered in patients with Crohn’s disease. Moreover, GSN is an actin-depolymerizing protein that regulates actin dynamics and is involved in cytoskeletal remodeling [16,17]. Its extracellular isoform, plasma GSN, is expressed in the blood, urine, and other extracellular fluids, such as lymph, burn wound fluid, cerebrospinal fluid, and airway surface fluids [19,20]. The secreted GSN also functions in the extracellular actin scavenger system, where it is responsible for the severance and removal of actin filaments from dead cells into the bloodstream [13]. In addition, the secreted GSN binds to lipopolysaccharides, which are compounds derived from the cell wall of Gram-negative bacilli, and inhibits the activation of Toll-like receptors, thereby regulating immune responses [21,22,23,24]. The secreted GSN has anti-inflammatory properties, and decreased GSN levels in the blood have been reported in chronic inflammatory diseases. Although the mechanism by which the GSN levels in the blood are reduced remains unclear, the re-distribution of GSN to inflammatory sites, binding to some plasma factors secreted in association with inflammation, and decreased GSN production have previously been reported [23,24]. Therefore, in this study, we examined whether GSN can be used as a biomarker for the clinical and endoscopic activities of UC.

## 2. Materials and Methods

### 2.1. Study Subjects and Sample Collection

In total, 138 patients with UC and 16 healthy controls were enrolled in this study at the Department of Gastroenterology and Hepatology, Nagoya University Hospital, between April 2016 and April 2021. The healthy controls comprised 10 women, and their median age was 45 (range 36–66) years. Patients were diagnosed with UC based on clinical, endoscopic, and histological criteria and received medical therapy. Clinical and endoscopic activity scores were reviewed from their medical records. Blood sampling and endoscopy were performed within a maximum interval of 1 month. Serum was obtained from the blood samples and stored at −80 °C until GSN analysis. Patients with UC comprised 84 women and 54 men, and their median age was 47 (range 20–82) years. The median duration of disease was 143 (range 7–372) months. The median C-reactive protein level was 0.08 (range 0–8.4) mg/dL. The median albumin level was 4.1 (range 1.8-4.9) g/dL. The median Mayo score was 3 (range 0–12). Here, 74.6% (103/138) patients were administered 5-aminosalicylic acids, 13% (18/138) patients were administered corticosteroids, and 32.6% (45/138) patients were administered biologic agents. Patients with UC were classified according to the extent of disease involvement as those with proctitis, left-sided colitis, or pancolitis, as described in the Montreal classification.

The proportion of patients with proctitis, left-sided colitis, and extensive colitis was 5.7% (8/138), 26% (36/138), and 68.3% (94/138), respectively. Clinical activity was determined using the Mayo score, and remission was defined by a score of ≤2. The endoscopic Mayo score was used to determine endoscopic activity, and endoscopic remission was defined by a score of 0. The proportion of clinically and endoscopically active patients was 56.5% (70/138) and 63.6% (88/138), respectively. The patient characteristics are presented in Table 1.

### 2.2. Measurement of Serum GSN Level

Serum GSN level was measured using an enzyme-linked immunosorbent assay (ELISA) kit (Abcam, Cambridge, UK), according to the manufacturer’s instructions. The absorbance of each sample was measured at 450 and 570 nm using a PowerScan4 microplate reader (DS Pharma Medical Co., Osaka, Japan). The level of GSN was calculated using a standard curve.

### 2.3. Mass Spectrometry

Lower rectum intestinal mucus samples of 3 patients with active UC and 3 patients with UC in remission were collected through colonoscopy. Colon mucus from the anterior and right rectal walls was collected using brush catheters (Colonoscope Cytology Brush^®^; Cook Medical, Winston-Salem, NC, USA).

Patients were diagnosed with UC based on clinical, endoscopic, and histological criteria. These samples were lysed using a Minute Total Protein Extraction Kit for mass spectrometry (Funakoshi, Tokyo, Japan), and the specimens were adjusted to the same protein level before mass spectrometry (MS).

The proteins were digested using trypsin for 16 h at 37 °C after reduction and alkylation, and the peptides were analyzed using LC−MS on an Orbitrap Fusion mass spectrometer (Thermo Fisher Scientific Inc., Waltham, MA, USA) coupled to an UltiMate3000 RSLC nano LC system (Dionex Co., Amsterdam, The Netherlands), using a nano HPLC capillary column (Nikkyo Technos Co., Tokyo, Japan) with a nano electrospray ion source. Reverse-phase chromatography was performed with a linear gradient (0 min, 5% B, 100 min, 40% B) of solvent A (2% acetonitrile with 0.1% formic acid) and solvent B (95% acetonitrile with 0.1% formic acid). A precursor ion scan was carried out using a 400–1600 mass to charge ratio (m/z) before MS/MS analysis.

### 2.4. Data Analysis

The raw data were processed using Proteome Discoverer 1.4 (Thermo Fisher Scientific) in conjunction with the MASCOT search engine, version 2.6.0 (Matrix Science Inc., Boston, MA, USA) for protein identification. The peptides and proteins were identified using the human protein database in UniProt (release 2020_03) with a precursor mass tolerance of 10 ppm and a fragment ion mass tolerance of 0.8 Da. Fixed modification was set to carbamidomethylation of cysteine, and variable modification was set to oxidation of methionine.

### 2.5. Statistical Analysis

All analyses were performed using Prism software (GraphPad prism version 8 Software, GraphPad Software, San Diego, CA, USA). Differences between groups were compared using Mann–Whitney U-test and Kruskal–Wallis test. The area under the receiver operating characteristic (ROC) curve (AUC) was calculated by plotting sensitivity on the *y* axis against 1—specificity on the *x* axis for each value. The correlation analysis was performed using Pearson coefficients. Statistical significance was defined as *p* < 0.05 (*, *p* < 0.05; **, *p* < 0.01; and ***, *p* < 0.001).

### 2.6. Ethical Considerations

This study was approved by the ethics committee of the Nagoya University Hospital, Japan (Protocol number 2015-0420, August 2016). Written informed consent was obtained from all patients before their enrollment in accordance with tenets of the Declaration of Helsinki.

## 3. Results

### 3.1. Downregulation of Serum GSN in Patients with Clinically Active UC

We first conducted proteomic analysis of the specimens from patients with UC in the active phase and in remission. The specimens were collected from the anterior and right walls of the rectal mucosa using brush samples. We identified 460 proteins with a score of ≥30 from the brush samples in patients with active UC (Appendix A). Inflammatory protein markers (protein S100-A9) and a neurotrophic protein (myeloperoxidase) presented high scores. We compared protein expression in patients with UC in remission and in the active phase. Consistent with previous study results, Mucin-5B and Mucin-13 were downregulated in active UC compared with those in remission UC [25,26]. In addition, we found that GSN was downregulated in patients with active UC compared with that in patients with remission UC (Table 2).

We then compared the serum GSN level between patients with UC and healthy controls. We analyzed samples from 138 patients (54 males and 84 females) whose median age was 47 (20–82) years. Of all patients, 68.1% had extensive colitis, 26% had left-sided colitis, and 5.8% had proctitis. The proportion of clinically and endoscopically active patients was 56.5% (70/138) and 63.6% (88/138), respectively. The serum GSN level was lower in patients with UC than in the healthy controls (138 patients with UC and 16 healthy controls, *p* < 0.001, Figure 1a). In addition, the serum GSN level was significantly lower in clinically active patients with UC than in those in remission (138 patients with UC, *p* < 0.001, Figure 1b). The correlation between the GSN levels and Mayo scores was determined using Pearson coefficients, and a significant correlation was found (r = −0.70229, *p* < 0.001) (Appendix A).

The expression of CRP, which is used as a serum biomarker for UC, is induced by IL-6, but the expression of GSN is downregulated by a mechanism different from that of CRP. The correlation between the sensitivity of CRP and disease activity is low; therefore, we tested whether GSN is effective for patients in whom activity was difficult to assess with CRP. Among the 82 patients with UC whose CRP level was normal (<0.14 mg/dL), the GSN level was significantly lower in patients with clinically active disease than in those in the remission phase (82 patients with UC, *p* < 0.001, Figure 1c). These findings indicate that the GSN level correlates with clinical activity, even in cases with a normal CRP level.

### 3.2. Inverse Relationship between the Serum GSN Level and Endoscopic Activity in Patients with UC

We analyzed whether the serum GSN level is associated with endoscopic activity in patients with UC. A decreased GSN level correlated with an increased endoscopic activity score (Mayo 0 vs. 1, *p* = 0.999; Mayo 0 vs. 2, *p* = 0.0113: Mayo 0 vs. 3, *p* < 0.01; Mayo 1 vs. 2, *p* = 0.0549; Mayo 1 vs. 3, *p* < 0.001; and Mayo 2 vs. 3, *p* < 0.001). Patients with MES 2 had a lower GSN level than those with MES 0 (54 patients (Mayo = 2), 50 patients (Mayo = 0), *p* = 0.0113), possibly reflecting minor mucosal changes (Figure 2a).

Recently, mucosal healing has been reported to reduce operative and relapse rates, and therapeutic goals have shifted from symptom relief to mucosal healing [3,4,5]. The detection of mucosal healing is important when using a serum biomarker that correlates with UC activity. We defined mucosal healing using the Mayo endoscopic score of 0 and tested whether mucosal healing could be detected by the GSN level.

The GSN level was lower in patients with endoscopically active UC (Mayo endoscopic score (MES) > 0) than in those with endoscopic remission (MES = 0) (*p* < 0.001, Figure 2a). The correlation between the GSN levels and Mayo endoscopic scores was measured using Pearson coefficients, and a significant correlation was found (r = −0.7585, *p* < 0.01) (Appendix A). Pearson coefficient was also used to determine the correlation between the GSN and CRP levels, and it was found that they had a low correlation (r = −0.287, *p* = 0.006) (Appendix A).

The correlation between the GSN levels and Mayo endoscopic scores was measured using Pearson coefficients, and it was found that the GSN level and albumin had a low correlation (r = 0.44755, *p* < 0.001) (Appendix A).

In addition, we tested whether the GSN level could detect mucosal healing in cases with a normal CRP level. Among the 82 patients with UC whose CRP level was within the normal level (<0.14 mg/dL), the GSN level was significantly lower in patients in the endoscopically active phase than in those in the remission phase (*p* < 0.001, Figure 2c). These findings indicate that the GSN level could detect clinical and endoscopic activities in UC patients with high sensitivity. Furthermore, even in patients with a normal CRP level, it correlated with clinical and endoscopic activities, making CRP useful for patients whose activity is difficult to assess with conventional blood tests.

### 3.3. GSN as a Serological Biomarker of Clinically and Endoscopically Active UC

Given that the GSN level correlated with clinical and endoscopic activities in patients with UC, we next investigated its diagnostic potential to detect clinical remission and mucosal healing in order to use it as a serum biomarker.

We compared the sensitivity and specificity of GSN with those of CRP using ROC curve and AUC analyses. The sensitivity and specificity of GSN were 91.43% and 89.71%, respectively, for the detection of clinical remission at a cut-off of 10.67 μg/mL (Figure 3a). The AUC of GSN was 0.874 and that of CRP was 0.78. For the detection of endoscopic remission, the sensitivity and specificity of GSN were 78.41% and 86.54%, respectively (Figure 3c), whereas those of CRP were 56.82% and 82.00%, respectively (Figure 3d). The AUC of GSN was 0.835, and that of CRP was 0.692. The AUC of GSN was higher than that of CRP for identifying both clinical and endoscopic remission (Figure 3a–d). These data suggest that GSN is a biomarker that reflects clinical and endoscopic activities and that it can detect mucosal healing.

To determine whether GSN can be used as a biomarker for assessing the clinical and endoscopic activities of UC, we analyzed its sensitivity and specificity using the ROC curve and AUC analyses and compared the results of GSN and CRP, an existing UC marker. The AUC of GSN was higher than that of CRP for identifying both clinical and endoscopic remission (Figure 3a–d).

## 4. Discussion

In this study, we showed that the GSN level correlates with the clinical and endoscopic activities of UC. GSN also showed high sensitivity and specificity in predicting the achievement of mucosal healing in patients with UC.

Currently, CRP and LRG are used as blood-based biomarkers to evaluate the activity of IBD. CRP expression is induced by IL-6 and is used to evaluate various inflammatory diseases [27]. CRP is a useful marker for the diagnosis of IBD, evaluation of disease activity, and prediction of therapeutic efficacy [8,9,28]. However, CRP correlates mostly with severe inflammation and does not reflect mild inflammation [29].

We performed proteomic analysis of the specimens from patients with UC in the active phase and remission phase. We identified 460 proteins in patients with active UC, as in previous studies, the inflammatory markers such as S100-A9 and myeloperoxitase were detected. In addition, IgG-Fc, which is required for the stabilization of Mucin-2 was also detected. Among them, we focused on GSN, the expression of which was downregulated in patients with active UC compared with that in patents with remission UC, and its expression in patents with remission UC was higher than that in patients with active UC.

The GSN level decreases with endoscopic activity, and patients with MES 2 have significantly lower GSN level than patients with MES 0, suggesting that GSN may also reflect mild intestinal inflammation. In addition, the fact that the GSN level reflects clinical and endoscopic activities even in a group of patients with normal CRP levels suggests that the GSN level may be useful for patients whose activity has been difficult to assess with conventional biomarkers. Furthermore, in this study, we demonstrated that the GSN level reflects clinical and endoscopic remission with higher sensitivity and specificity than CRP. We believe that GSN can help detect IL-6-independent inflammation and mucosal healing because it reflects even mild inflammation. Moreover, it can be used to assess endoscopic activity even in CRP-negative cases and could be a new biomarker with an underlying mechanism of action that is different from that of CRP. LRG, a newly identified serum marker for UC, has also been reported to correlate with clinical and endoscopic activities of the disease [10,12]. In the future, it will be necessary to compare the sensitivity and specificity of LRG and GSN, and utilize them according to disease activity and stage. Furthermore, it has been suggested that the measurement of both LRG and GSN may allow more accurate assessment of disease activity and predict mucosal healing with higher sensitivity and specificity.

Mucosal healing was previously defined as MES 0 and MES 1; however, as patients with MES 1 have a higher relapse rate than those with MES 0 [4,30], several studies have considered only MES 0 to reflect mucosal healing. Therefore, mucosal healing was defined as MES 0 in this study. As the operation and relapse rates are low in patients who have achieved mucosal healing, mucosal healing has become the therapeutic goal in UC. GSN presented higher sensitivity and specificity than CRP in detecting mucosal healing with a cut off of 10.67 g/mL. UC is a chronic inflammatory disease with recurrent remissions and relapses, and optimization of treatment based on more accurate assessment of disease activity is needed. Optimal therapeutic options may improve the prognosis of patients by enabling long-term maintenance of mucosal healing. Therefore, using GSN as a biomarker will enable accurate assessment of mucosal healing and treatment optimization.

GSN is a multifunctional protein with altered blood levels in chronic inflammatory and autoimmune disorders such as rheumatoid arthritis [17,31], ankylosing spondylitis [32], systemic lupus erythematosus [33], and Henoch–Schoenlein purpura [34]. However, there have been no reports on the association between GSN level and disease activity in patients with IBD. In the gastrointestinal tract, GSN, along with the GSN superfamily protein villin-1, regulates actin dynamics, intestinal epithelial cell death, and intestinal inflammation [35], but its function in IBD is unknown, and the mechanism of its decreased expression in the intestinal tissues and blood requires further analysis. GSN binds to lipopolysaccharides (LPS), a bacterial cell wall component, and inhibits the activation of Toll-like receptors on the surface of innate immune system cells, such as macrophages and dendritic cells. In IBD, the intestinal epithelial barrier, including mucus production and tight junction formation, is disrupted, and this disruption induces bacterial translocation of LPS from the intestinal tract into the bloodstream. The progressive disruption of the intestinal epithelial barrier mechanism associated with inflammation in IBD may induce an increase in the blood levels of LPS and decrease the level of GSN. In addition, GSN has been reported to be associated with multiple immune cell functions, such as neutrophil migration, suggesting that the abnormal activation of immune cells associated with chronic inflammation may be related to the mechanism of decreased GSN level. The proteins that we identified using proteomic analysis included calprotectin, which is currently used as a stool marker, and could have comprised proteins that can be used as blood- or stool-based markers of disease activity.

Our study had some limitations. For instance, it was a retrospective, single-center study with a small number of patients with heterogeneous backgrounds and treatments. Future studies should involve the recruitment of a prospective cohort to ascertain whether GSN reflects endoscopic activity and mucosal healing in patients with UC. As mentioned earlier, GSN is affected by other inflammatory and autoimmune disorders, and therefore, it may not be useful when other inflammations are involved. Prospective correlations between the GSN level and clinical and endoscopic activities should be carefully examined for the presence of intestinal and other infections or other autoimmune complications.

Nevertheless, we believe that GSN has the potential to be developed into a biomarker to assess UC disease activity and mucosal healing, and can contribute to the realization of treatment targets aimed at achieving mucosal healing. Our findings may lead to a reduction in the number of endoscopic procedures that are needed to assess UC disease activity, reducing patient stress and medical costs. Furthermore, non-invasive markers for disease activity will enable us to accurately assess UC disease activity and adjust treatments appropriately, as well as enable the use of treat-to-target approaches to achieve mucosal healing.

## 5. Patents

This work has been submitted for a patent application.

## Figures and Tables

**Figure 1 biomedicines-10-00872-f001:**
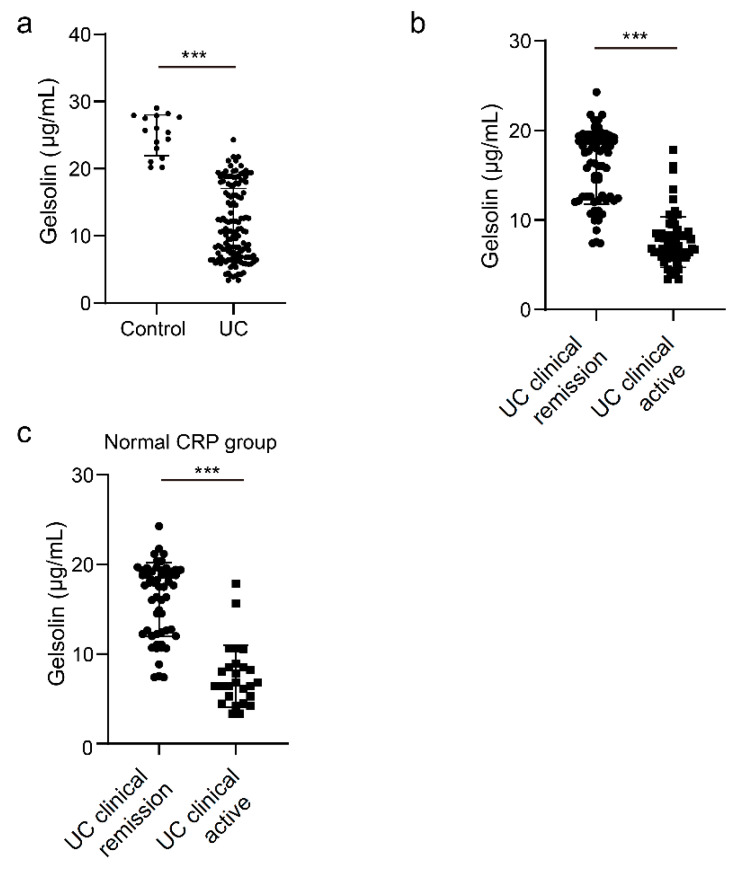
Serum gelsolin (GSN) level decreased in clinically active patients with ulcerative colitis (UC). Serum GSN level was measured in (**a**) 138 patients with UC and 16 healthy subjects (control); (**b**) 68 patients with UC in clinical remission and 70 patients with clinically active UC; and (**c**) 56 patients with UC in clinical remission and 26 patients with clinically active UC and normal C-reactive protein (CRP) level (<0.14 mg/dL). Statistical significance was defined as *p* < 0.05 (*** *p* < 0.001) using Mann–Whitney U-test.

**Figure 2 biomedicines-10-00872-f002:**
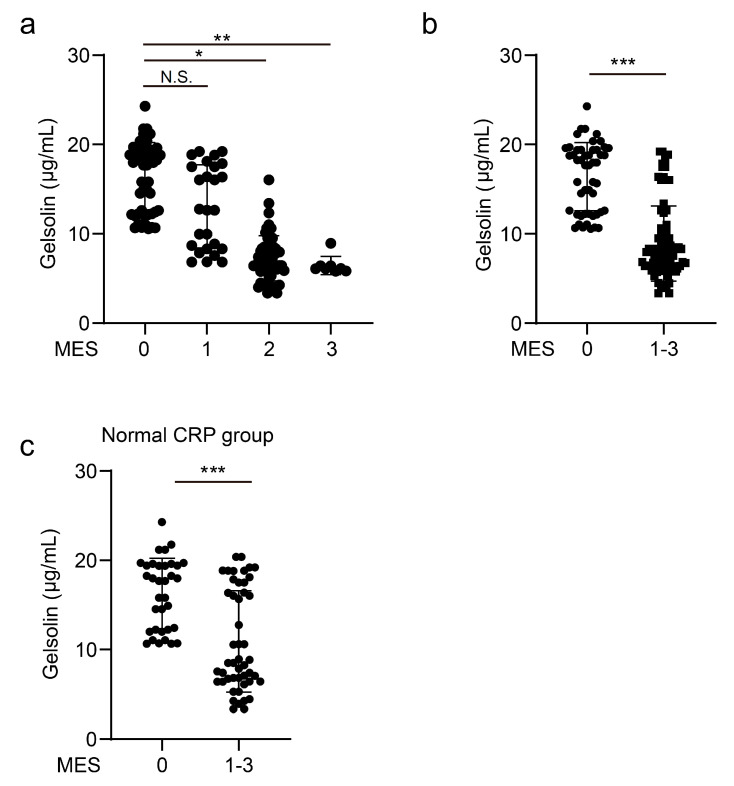
GSN level correlates with the endoscopic activity score in patients with UC. (**a**) Serum GSN level in patients with UC categorized according to disease activity (MES 0 (*n* = 50), 1 (*n* = 26), 2 (*n* = 54), and 3 (*n* = 8)). (**b**) Serum GSN level in 50 patients with UC in endoscopic remission (Mayo endoscopic score (MES) = 0) and 88 patients with endoscopically active UC (MES > 0). (**c**) Serum GSN level was measured in 34 patients with UC in endoscopic remission and 48 patients with endoscopically active UC and normal CRP level (CRP < 0.14 mg/dL). Statistical significance was defined as *p* < 0.05 (* *p* < 0.05; ** *p* < 0.01; *** *p* < 0.001; and N.S., not significant) using Mann–Whitney U-test and Kruskal–Wallis test.

**Figure 3 biomedicines-10-00872-f003:**
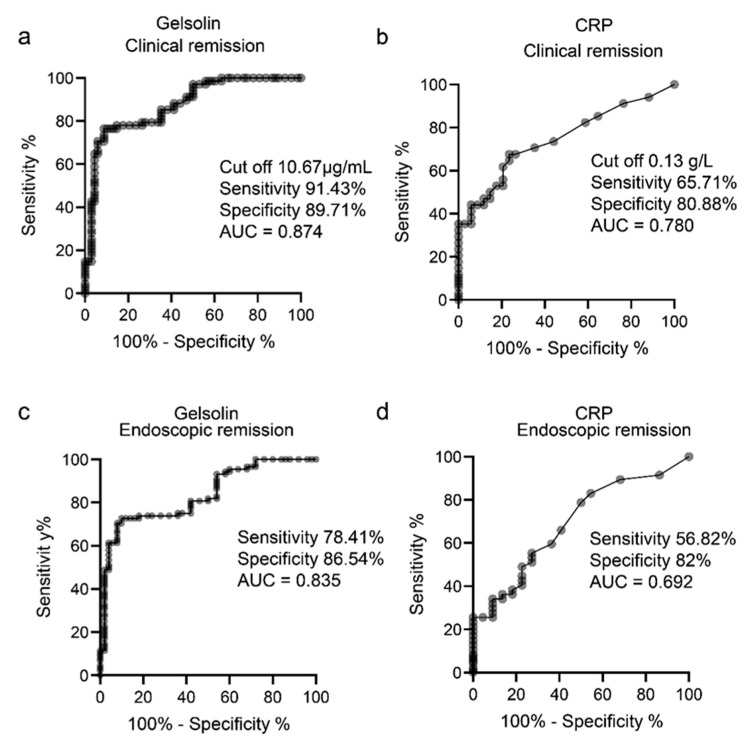
GSN level reflects clinical and endoscopic activities in patients with UC. Receiver operating characteristic (ROC) curves for GSN and CRP indicating their sensitivity and specificity in discriminating (**a**,**b**) clinical remission and (**c**,**d**) endoscopic remission.

**Table 1 biomedicines-10-00872-t001:** Characteristics of patients with ulcerative colitis.

Patient Characteristic	*N* = 138
Sex, female/male, *N*	84/54
Age, years, median (range)	47 (20–82)
Duration of disease, months (range)	143 (7–372)
Disease location, *N* Extensive/left-sided/proctitis	94/36/8
Treatment, *N* Oral 5-aminosalicylic acid, *N* (%) Corticosteroids, *N* (%) Biologic agents, *N* (%) Immunomodulators, *N* (%) Calcineurin inhibitors, *N* (%) Topical agents, *N* (%)	103 (74.6) 18 (13.0) 45 (32.6) 27 (19.6) 3 (2.2) 45 (32.6)
C-reactive protein, mg/dL, median (range)	0.08 (0–8.4)
Albumin, g/dL, median (range)	4.1 (1.8–4.9)
Mayo score median (range)	3 (0–12)

**Table 2 biomedicines-10-00872-t002:** List of genes downregulated in active UC compared with those in remission UC.

Accession	Description	*p*
Q9HC84	Mucin-5B	0.03349
Q9H3R2	Mucin-13	0.004309
P06396	Gelsolin	0.03639

## Data Availability

The original data and the materials generated in this study are available from the corresponding author on request.

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
