# Peer review of "Gelsolin as a Potential Biomarker for Endoscopic Activity and Mucosal Healing in Ulcerative Colitis"

_biomedicines, 2022, doi:10.3390/biomedicines10040872_

Round 1

Reviewer 1 Report

The aim is stated clear. The authors stated clearly what study found and how they did it.

The title is informative and relevant.

The research question is clearly outlined. The research question also justified given what is already known about the topic. The process of selection of the subjects was clear. The variables are well defined and measured appropriately. The study methods are valid and reliable. There are enough details provided in order to replicate the study.

The data is presented in an appropriate way. The text in the results add to the data and it is not repetitive. Statistically significant results are clear. It is clear which results are with practical meaning. Results are discussed from different angles and placed into context without being overinterpreted.

The conclusions answer the aim of the study. The conclusions are supported by references and own results.

The limitations of the study are not fatal, but they are opportunities to inform future research.

Specific comments on weaknesses of the article and what could be improved:

Major points - none

Minor points

1. Did you perform some correlation analysis measured by Pearson coeff.?

Author Response

We thank you for the critical comments and valuable suggestions that have helped to improve our manuscript. Our responses to the comments are provided below. We have revised the manuscript according to the comments; the revised portions have been underlined. We have included the additional data as Figure S1–S3. We have added a description of the mechanisms of reduced gelsolin level in the main text. The revised manuscript was proofread by an English proofreading service provider.

1, Did you perform some correlation analysis measured by Pearson coeff.?

We performed further analysis of correlation between the GSN level and Mayo score or Mayo endoscopic score using Pearson coefficients. We found that the GSN level and Mayo score or Mayo endoscopic score had a significant correlation (Figures S1 and S2).

We have added these findings in the main text on page 4, lines 173–175:

“The correlation between the GSN levels and Mayo scores was determined using Pearson coefficients, and a significant correlation was found (r = -0.70229, P < 0.001) (Figure S1).”

And on page 6, lines 210–212:

“The correlation between the GSN levels and Mayo endoscopic scores was measured using Pearson coefficients, and a significant correlation was found (r = -0.7585, P < 0.01) (Figure S2).”

Please see the document.

Reviewer 2 Report

In the current manuscript “Gelsolin as a potential biomarker for endoscopic activity and mucosal healing in ulcerative colitis” by Maeda et al. showed usefulness of gelsolin as a serological biomarker for clinical and endoscopic activities in ulcerative colitis.

Comments:

  1. The protein levels of gelsolin was shown by mass spectroscopy. It would be interesting to detect and compare gelsolin levels by western blot and PCR from colonic samples from healthy and UC patients if given the availability of the few samples.
  2. Authors need to describe in introduction/discussion section for mechanisms behind the reduced gelsolin levels and its implications in UC patients by citing the existing literature.
  3. Does the levels of gelsolin correlates with any other parameters mentioned in Table 1, indicate in the table or in the results section.

Author Response

We thank you for the critical comments and valuable suggestions that have helped to improve our manuscript. Our responses to the comments are provided below. We have revised the manuscript according to the comments; the revised portions have been underlined. We have included the additional data as Figure S1–S3. We have added a description of the mechanisms of reduced gelsolin level in the main text. The revised manuscript was proofread by an English proofreading service provider.

1. The protein levels of gelsolin was shown by mass spectroscopy. It would be interesting to detect and compare gelsolin levels by western blot and PCR from colonic samples from healthy and UC patients if given the availability of the few samples.

As per your suggestion, we performed PCR analysis of the colonic samples from healthy individuals (n = 3) and patients with UC (n = 3) and found that GSN expression was lower in the colonic samples of the patients with active UC than in those of the healthy controls.

We wish to further validate the data, shown in the figure below, with a higher number of specimens; therefore, we refrain from including the results in the manuscript.

2. Authors need to describe in introduction/discussion section for mechanisms behind the reduced gelsolin levels and its implications in UC patients by citing the existing literature.

We have added the relevant description in the introduction and discussion sections.

“The secreted GSN has anti-inflammatory properties, and decreased GSN levels in the blood have been reported in chronic inflammatory diseases. Although the mechanism by which the GSN levels in the blood are reduced remains unclear, the re-distribution of GSN to inflammatory sites, binding to some plasma factors secreted in association with inflammation, and decreased GSN production have previously been reported.” (Page 2, lines 75 –81)

“GSN binds to lipopolysaccharides (LPS), a bacterial cell wall component, and inhibits the activation of toll-like receptors on the surface of innate immune system cells, such as macrophages and dendritic cells. In IBD, the intestinal epithelial barrier, including mucus production and tight junction formation, is disrupted, and this disruption induces bacterial translocation of LPS from the intestinal tract into the bloodstream. The progressive disruption of the intestinal epithelial barrier mechanism associated with inflammation in IBD may induce an increase in the blood levels of LPS and decrease in the level of GSN. Besides, GSN has been reported to be associated with multiple immune cell functions, such as neutrophil migration, suggesting that the abnormal activation of immune cells associated with chronic inflammation may be related to the mechanism of decreased GSN level.” (Page 9, lines 311–313, Page 10, lines 314-321).

3. Does the levels of gelsolin correlates with any other parameters mentioned in Table 1, indicate in the table or in the results section.

We appreciate the critical suggestion that improved our manuscript. Per your suggestion, we performed further analysis of the correlation between the GSN level and other parameters mentioned in Table 1 using Pearson coefficients. We found that the GSN level and CRP or Alb had a low correlation (Figure S3).

We have incorporated these findings in the main text on page 6, lines 212–217, as follows:

“Pearson coefficient was also used to determine the correlation between the GSN and CRP levels, and it was found that hey had a low correlation (r = -0.287, P = 0.006) (Figure S3A)

The correlation between the GSN levels and Mayo endoscopic scores was measured using Pearson coefficients, and it was found that the GSN level and albumin had a low correlation (r = 0.44755, P < 0.001) (Figure S3B).

Please see the document.
